# Genomic Prediction of Rust Resistance in Tetraploid Wheat under Field and Controlled Environment Conditions

**Shiva Azizinia [1,\*], Harbans Bariana [2], James Kolmer [3], Raj Pasam [1], Sridhar Bhavani [4], Mumta Chhetri [2], Arvinder Toor [2], Hanif Miah [2], Matthew J. Hayden [1,5], Dunia Pino del Carpio [1], Urmil Bansal [2,†] and Hans D. Daetwyler [1,5,†]**

[1] Agriculture Victoria, AgriBio, Centre for AgriBioscience, Bundoora, VIC 3083, Australia; raj.pasam@agriculture.vic.gov.au (R.P.); matthew.hayden@agriculture.vic.gov.au (M.J.H.); duniapc77@gmail.com (D.P.d.C.); hans.daetwyler@agriculture.vic.gov.au (H.D.D.)

[2] Plant Breeding Institute, School of Life and Environmental Sciences, Faculty of Science, The University of Sydney, 107 Cobbitty Road, Cobbitty, NSW 2570, Australia; harbans.bariana@sydney.edu.au (H.B.); Mumta.Chhetri@csiro.au (M.C.); arvi2006leo@gmail.com (A.T.); hanif.miah@sydney.edu.au (H.M.); urmil.bansal@sydney.edu.au (U.B.)

[3] United States Department of Agriculture–Agricultural Research Service (USDA-ARS), Cereal Disease Laboratory, St. Paul, MN 55108, USA; Jim.Kolmer@ars.usda.gov

[4] International Maize and Wheat Improvement Center (CIMMYT), 56237 Texcoco, Mexico; S.Bhavani@cgiar.org

[5] School of Applied Systems Biology, La Trobe University, Bundoora, VIC 3083, Australia

[\*] Correspondence: shiva.azizinia@agriculture.vic.gov.au

[†] Both authors contributed equally to this work.

**Abstract:** Genomic selection can increase the rate of genetic gain in crops through accumulation of positive alleles and reduce phenotyping costs by shortening the breeding cycle time. We performed genomic prediction for resistance to wheat rusts in tetraploid wheat accessions using three cross-validation with the objective of predicting: (1) rust resistance when individuals are not tested in all environments/locations, (2) the performance of lines across years, and (3) adult plant resistance (APR) of lines with bivariate models. The rationale for the latter is that seedling assays are faster and could increase prediction accuracy for APR. Predictions were derived from adult plant and seedling responses for leaf rust (Lr), stem rust (Sr) and stripe rust (Yr) in a panel of 391 accessions grown across multiple years and locations and genotyped using 16,483 single nucleotide polymorphisms. Different Bayesian models and genomic best linear unbiased prediction yielded similar accuracies for all traits. Site and year prediction accuracies for Lr and Yr ranged between 0.56–0.71 for Lr and 0.51–0.56 for Yr. While prediction accuracy for Sr was variable across different sites, accuracies for Yr were similar across different years and sites. The changes in accuracies can reflect higher genotype × environment (G × E) interactions due to climate or pathogenic variation. The use of seedling assays in genomic prediction was underscored by significant positive genetic correlations between all stage resistance (ASR) and APR (Lr: 0.45, Sr: 0.65, Yr: 0.50). Incorporating seedling phenotypes in the bivariate genomic approach increased prediction accuracy for all three rust diseases. Our work suggests that the underlying plant-host response to pathogens in the field and greenhouse screens is genetically correlated, but likely highly polygenic and therefore difficult to detect at the individual gene level. Overall, genomic prediction accuracies were in the range suitable for selection in early generations of the breeding cycle.

**Keywords:** tetraploid wheat; genomic selection; multivariate analysis; adult plant resistance; all stage resistance

## 1. Introduction

Tetraploid wheat has played important role in common wheat evolution and thus human history and comprises 5% of total global wheat production [1]. With origins tracing to Middle East including the current territories of Turkey, Syria, Iraq and Iran, durum wheat is more adapted to the dry Mediterranean climate [2] and is known for its hardness, protein, nutty flavour and a range of products.

Stem rust (Sr, caused by *Puccinia graminis* f. sp. *tritici* Eriks; Pgt), leaf rust (Lr, *P. triticina* Eriks; Pt) and stripe rust (Yr, *P. striiformis* f. sp. *tritici* Westend.; Pst) pose a major threat to global production of durum wheat. These three rust diseases can cause substantial losses in yield and quality, however, the losses due to Sr can be up to 100%, where susceptible varieties are grown [3]. To date, the most economic and environment-friendly approach to control wheat rust diseases is to pyramid two or more rust resistance genes in new cultivars [4].

Rust resistance has often been categorised as all stage resistance (ASR) and adult plant resistance (APR). ASR is controlled either by single or a few genes of large effects, whereas an acceptable level of APR is often controlled by a combination of multiple minor-effect resistances genes [5,6]. While promising progress has been made on the identification of ASR genes and shortening of breeding cycles, discovery of APR genes still lags behinds. Therefore, detection of APR genes in the presence of ASR genes in breeding populations remains a challenge.

The use of genome wide markers and genomic selection has been proposed to overcome such issues [7]. Genomic selection (GS) has the potential to reduce cost of phenotypic evaluations in plants and shorten the breeding cycle. Using genomic prediction models, genotypic information and phenotypic data for various traits from training populations, are used to calculate marker effects and estimate the genetic estimated breeding value (GEBV) of a candidate. The implementation of GS depends on several factors, among which the most important ones are model performance, sample size, marker density, heritability, population structure, trait genetic architecture, as well as training and breeding population relatedness [8–10].

A variety of genomic prediction statistical models have been tested in many crops for traits with different genetic architectures. The most commonly used genomic prediction models with high predictive accuracy are linear parametric methods such as ridge-regression best linear unbiased prediction (RR-BLUP), Bayes A, B, Cπ, Bayesian Lasso, non-linear semi-parametric methods such as; reproducing kernel Hilbert space (RKHS), non-linear, non-parametric methods (Random Forest; RF) and linear non-parametric methods (Support vector machine; SVM) [11,12]. The prediction accuracy of these models depends on the interplay of trait genetic architecture and population diversity [13,14].

One of the major benefits of applying GS for quantitative disease resistance breeding is that more candidates can be evaluated earlier in the breeding cycle, leading to higher selection intensity. Decreasing the breeding cycle time through rapid generation advancement can significantly increase rates of genetic gain through accumulation of positive alleles that condition economic traits. GS can be applied to select single plants in early generations or among lines prior to advanced testing for all traits of interest [15]. Another advantage is quantitative resistance can be predicted in any individual or line regardless of whether R genes are present [16].

Rust resistance in wheat is possibly one of the best-studied patho-systems for using GS models in disease resistance breeding [17–24]. Rutkoski et al. [17] proposed a GS-based wheat breeding scheme for quantitative resistance to stem rust in wheat that could reduce the breeding cycle time by up to two-fold while facilitating the pyramiding of ASR and APR genes.

The aim of this study was to estimate the accuracy of genomic prediction for rust resistance in a diverse set of tetraploid wheat accessions from the Watkins Collection [25]. ASR and APR

data that were collected across several years and sites was used for various prediction models in a cross-validation schemes. The effect of using ASR seedling data in predicting adult plant response in a bivariate model was explored.

## 2. Materials and Methods

### 2.1. Plant Materials

Plant material consisting of 391 tetraploid wheat accessions (344 *Triticum durum; 45 T. turgidum; 15 T. dicoccon*) with diverse origins, derived from a collection assembled by AE Watkins comprising of 1284 wheat accessions [25]. The geographic distribution of the accessions covered regions from Africa to the Middle East cover 30 countries (Figure 1).

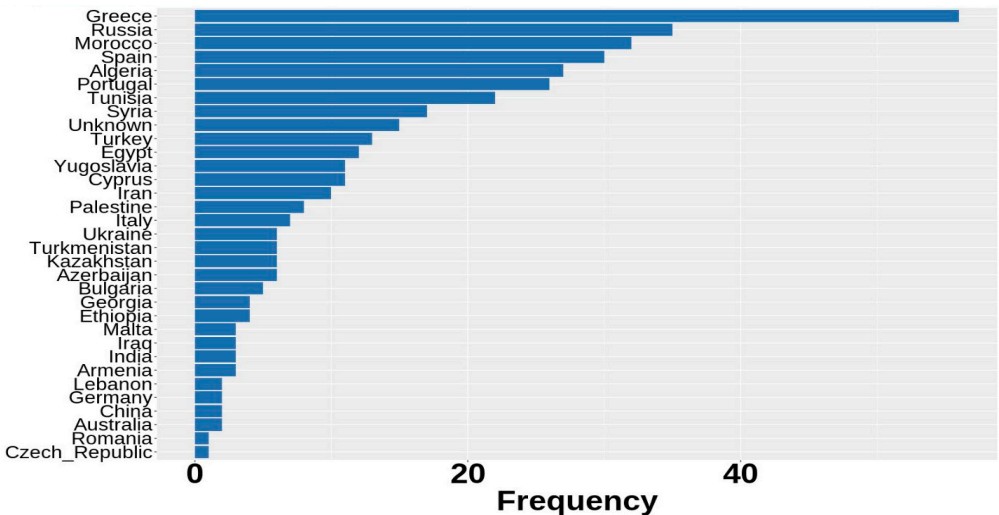

**Figure 1.** Geographical distribution of accessions used in the current study.

### 2.2. Seedling and Field Evaluations

Field evaluation of the tetraploid wheat panel at the adult-plant stage was carried out across several years (2005, 2006, 2012 and 2013) and locations including the Ethiopian Institute for Agricultural Research Center in Debre Zeit (EIAR-DZ) and the Lansdowne (LD), Karalee (KL), Richmond (RM) and Horse Unit (HU) sites of the University of Sydney Plant Breeding Institute in Australia (Table 1).

The EIAR-DZ is a hotspot for evaluating durum-specific Pgt races. The plant material was planted in hill plots with 15–20 seeds/line. Each block of four experimental rows was surrounded by a single row each side vertically and 30 cm strip of susceptible genotype horizontally to facilitate good rust epidemics. Stem rust spreaders of susceptible wheat cultivars were artificially inoculated 2–3 times with races TTKSK (Ug99) and JRCQC starting from stem elongation stage along with other races such as TKTTF, TRTTF, RRTTF known to be present in the nursery [26]. Stem rust disease severity and infection type responses were assessed twice at weekly intervals at soft dough and early grain fill stages of plant development. Disease severities were scored following a modified Cobb scale [27,28]. The lines were classified, based on the host response into resistant (R), moderately resistant (MR), intermediate (M), moderately susceptible (MS), and susceptible (S) as described by Roelfs et al. [28].

The panel was also tested at the University of Sydney Plant Breeding Institute, Cobbitty the same as at EIAR-DZ against commercially important races of all three rust pathogens under irrigated conditions according to Randhawa [29].

**Table 1.** Number of observation (N), number of replicates (R) and average resistance response (±) standard deviations for leaf rust (Lr), stem rust (Sr) and stipe rust (Yr).

|  | Trait | Year | Site | R | N | Mean Response |
|---|---|---|---|---|---|---|
| Field | Lr | 2005 | Lansdowne | 1 | 376 | 3.45 ± 1.95 |
|  |  | 2005 | Richmond | 1 | 195 | 3.66 ± 1.87 |
|  |  | 2006 | Lansdowne | 2 | 388/355 | 2.73 ± 1.62 |
|  |  | 2013 | HorseUnit | 1 | 307 | 3.62 ± 1.52 |
|  | Sr | 2005 | Lansdowne | 1 | 388 | 4.90 ± 2.14 |
|  |  | 2006 | Lansdowne | 1 | 383 | 4.84 ± 1.85 |
|  |  | 2012 | Karalee | 1 | 331 | 5.07 ± 1.71 |
|  |  | 2012 | Ethiopia | 1 | 350 | 4.59 ± 1.39 |
|  |  | 2013 | Horse Unit | 1 | 310 | 4.81 ± 1.42 |
|  | Yr | 2005 | Lansdowne | 1 | 389 | 3.68 ± 1.15 |
|  |  | 2006 | Lansdowne | 2 | 390 | 3.89 ± 1.46 |
|  |  | 2012 | Karalee | 1 | 334 | 2.98 ± 1.03 |
| Greenhouse | Lr | 2012 | USA | 3 | 377 | 5.93 ± 2.32 |
|  | Sr | 2013 | AUS | 10 | 174 | 6.78 ± 1.69 |
|  | Yr | 2013 | AUS | 10 | 391 | 6.09 ± 2.29 |

Seedling evaluations for leaf rust was performed at the Cereal Disease Laboratory (CDL), Minnesota with three P. triticina (Pt) races (CA 1.2, 09 AZ 103A and THBJ). The panel was planted in 3.5 cm$^2$ plastic pots, with 5–6 seeds/genotype and 4 genotypes per pot planted in the quarterly corners that were filled with vermiculite and placed in plastic trays, six pots per tray. When the primary leaves were fully expanded genotypes were inoculated with Pt races. For seedling inoculations about 1 mg of urediniospores were mixed with 0.2 mL of Soltrol 170 oil. After inoculation, seedlings were allowed to dry for 1 h and then placed in a mist chamber overnight at 18 °C and 100% relative humidity. After incubation the seedlings were placed on a greenhouse bench with supplemental metal halide lighting (temperature ranging from 18 °C to 23 °C). Seedlings were fertilized with a 20-20-20 NPK solution immediately after inoculation and at 14 days after planting. The infection types (ITs) on the primary leaves of individual plants were recorded 10 days after inoculation using a 0–4 scale detailed in Long and Kolmer [30].

The panel was further tested at the University of Sydney Plant Breeding Institute, (PBI), Cobbitty under greenhouse conditions against nine Pgt races at the two-leaf stage according to McIntosh et al. [31]. The inoculated seedlings were incubated for 48 h at 18–20 °C in water filled steel trays covered with clear plastic hoods to maintain high humidity under natural light conditions. Following incubation these were shifted to temperature and irrigation-controlled microclimate rooms maintained at 25 °C. Infection was allowed to develop for 14 days. Seedlings responses were scored on a 0–4 scale [31].

For stripe rust, inoculations were carried out on about 12 days old seedlings. Urediniospores suspended in light mineral oil (Isopar L®) (5 mg spores per 10 mL of oil for 200 pots) were sprayed with a hydrocarbon propellant pressure pack sprayer. The inoculation equipment was thoroughly washed with 70% alcohol and rinsed under tap water to prevent any contamination between two successive inoculations. Inoculated seedlings were kept overnight in a room at 9–12 °C on trolleys filled with water and covered with plastic hoods to generate high levels of humidity. The material was then transferred to greenhouse microclimate room maintained at 17–18 °C. Stripe rust responses were scored after 14 days on a 0 - 4 scale devised by Gassner and Straib in 1932 (cited from [31]), with some modifications suggested by Wellings et al. [32], where 0 = immune and 4 = very susceptible.

*2.3. Genotyping*

Genomic DNA was extracted from bulked leaf material from 6–8 seedlings of each line. An Illumina iSelect genotyping array of 90 K gene-associated single nucleotide polymorphisms (SNPs) [33] was used

to genotype the tetraploid wheat accessions. SNP genotype calling was performed as described in Maccaferri et al. [34]. Data were filtered to remove SNPs with >30% missing data and a minor allele frequency of <1%. Missing data for the remaining SNPs was imputed using software Linkimpute [35]. A total of 16,483 SNPs with missing data imputed were used for further analysis. Among all the SNPs a total of 11,882 were unambiguously assigned to specific chromosomes and genetic map positions, the remaining 4601 SNPs were not assigned to chromosomes.

### 2.4. Population Structure

The accuracy of genomic prediction can be affected by population structure [36], which was investigated using principal component analysis. Applying all 16,483 SNPs, a genomic relationship matrix (**G**) was calculated using the R package sommer [37] and principal components were extracted from the **G** matrix using the R function prcomp [38]. Data were plotted to visualize population groupings in the collection using the R function ggplot2 [39]. A heat map with increasing colour, indicating higher relatedness of accessions was plotted.

### 2.5. Genomic Prediction Models

#### 2.5.1. Single Trait Genomic Model

Genomic Best Linear Unbiased Prediction (G-BLUP), a mixed linear model, is a modification of the conventional BLUP model [40], where genomic-relationships are used instead of the conventional pedigree-relationships. G-BLUP fits individuals as random effects, and the covariance among individuals is given by **G** estimated from genome-wide markers. This uses information from relatives for breeding value prediction, with closer relative information weighted more heavily. Linear models assume that the trait is conditioned by an infinitesimal number of additive loci (no major genes). Although not true, this foundational assumption of quantitative genetics principles has produced good genomic prediction results. However, in traits in which there are large-effect loci, assuming unique marker variances, non-linear (Bayesian) predictions can lead to better accuracies [41]. Here we used following basic linear model:

$$\mathbf{y} = \mathbf{1}\mu + \mathbf{Xb} + \mathbf{Zu} + \varepsilon \quad \mathbf{u} \;\sim\; N\left(0, \mathbf{G}\sigma_g^2\right), \tag{1}$$

where **y** is the vector of the response phenotypic trait, $\mu$ is the overall mean, **b** is the vector of fixed effects (interaction of year and site), **X** is the design matrix for **b**, **u** is the vector of random genetic effects (genomic breeding values), $\sigma_g^2$ is the variance captured by the SNP, **Z** is the design matrix for **u**, $\varepsilon$ is the vector of independent residuals distributed as $\varepsilon \sim N\left(0, \mathbf{I}\sigma_\varepsilon^2\right)$, and $\sigma_\varepsilon^2$ is the residual variance.

Best linear unbiased estimates (BLUE) were calculated using rust scores, implementing Model 1, where accessions were considered as fixed effects. In field experiments years and sites were used as fixed effects to adjust phenotypes, while pathotypes were considered as fixed effects in greenhouse experiments. BLUEs were used for calculating the accuracy of genomic prediction as correlation with GEBVs.

#### 2.5.2. Bi-Variate Genomic Model

Bivariate genomic breeding values were predicted as:

$$\begin{bmatrix} \mathbf{y}_1 \\ \mathbf{y}_2 \end{bmatrix} = \begin{bmatrix} \mathbf{I}_1 & 0 \\ 0 & \mathbf{I}_2 \end{bmatrix} \begin{bmatrix} \mu_1 \\ \mu_2 \end{bmatrix} + \begin{bmatrix} \mathbf{Z}_1 & 0 \\ 0 & \mathbf{Z}_2 \end{bmatrix} \begin{bmatrix} g_1 \\ g_2 \end{bmatrix} + \begin{bmatrix} e_1 \\ e_2 \end{bmatrix}, \tag{2}$$

where $\begin{bmatrix} \mathbf{y}_1 \\ \mathbf{y}_2 \end{bmatrix}$ is the vector of response variables of traits 1 and 2; $\mathbf{I}_1$ and $\mathbf{I}_2$ are identity matrices, $\begin{bmatrix} \mu_1 \\ \mu_2 \end{bmatrix}$ is the vector of intercepts of traits 1 and 2, $\begin{bmatrix} g_1 \\ g_2 \end{bmatrix}$ is the vector of genomic breeding values of the two traits,

$\mathbf{Z}_1$ and $\mathbf{Z}_2$ are the design matrices that associate genomic breeding values with response variables and $\begin{bmatrix} e_1 \\ e_2 \end{bmatrix}$ is the vector of random residuals of the two traits. It is assumed that $\begin{bmatrix} g_1 \\ g_2 \end{bmatrix} \sim N(0, \boldsymbol{I} \otimes \boldsymbol{H})$, where $\mathbf{H} = \begin{bmatrix} \sigma_{g1}^2 & \sigma_{g12} \\ \sigma_{g12} & \sigma_{g2}^2 \end{bmatrix}$ is the variance and covariance matrix of the genomic breeding values of the two traits and $\begin{bmatrix} e_1 \\ e_2 \end{bmatrix} \sim N(0, \boldsymbol{I} \otimes \boldsymbol{R})$, where $\mathbf{R} = \begin{bmatrix} \sigma_{e1}^2 & \sigma_{e12} \\ \sigma_{e12} & \sigma_{e2}^2 \end{bmatrix}$ is the residual variance and covariance matrix of the two traits. Breeding values for individual's responses in the seedling or adult plant stage were predicted from a multi-trait model including ASR and APR responses in the model as second trait in ASReml [42] in which an unstructured covariance matrix among traits was assumed.

### 2.5.3. Bayesian Models

In Bayesian models, the genetic variance was assumed to be non-equal across chromosomes or markers due to existence of major genes on some chromosomes. Because Bayesian models use a different approach for parameter estimation compared with mixed linear models, they can estimate unique marker variances.

### BayesA

With BayesA, each marker is assumed to have a unique variance. In this method each marker effect has a univariate normal prior, with mean zero and locus-specific variance and each locus may have distinct variance. The unconditional distributions of the marker effects follow identical and independent univariate t distributions, each with mean zero [7].

### BayesB

BayesA fits all markers in the model even if they have no effect on the trait of study. BayesB is an extension of BayesA and allows some markers to have no effect. BayesB employs a mixture distribution that includes a point of mass at zero and a univariate scaled t distribution. This method considers a proportion $\pi$ of markers with zero and $(1 - \pi)$ with non-zero effect. The parameter $\pi$ is always treated as a constant, which depends on the actual distribution of locus effects in real data analyses [7].

### BayesC

The assumption of BayesC is that each marker effect is zero with probability $\pi$ and follows a univariate normal distribution with probability $(1 - \pi)$ [43]. In probability $\pi = 0$, BayesC is equal to RR-BLUP.

### Bayes LASSO

In Bayesian LASSO, the least absolute shrinkage and selection operator (LASSO), which has been developed based on a regression method [44], markers are assumed to have an equal variance. In this method shrinkage of marker effects is more severe on small-effect markers, whereas larger-effect loci are shrunk less. These models essentially lead to variable selection, as some loci are estimated to have near zero effect.

### Reproducing Kernel Hilbert Spaces

RKHS is an alternative for multiple linear regression to capture complex interaction patterns, which may be difficult to account for in linear models [45] and non-additive effects because of its ability to perform regression in a higher dimensional space.

## 2.6. Heritability and Correlations

Variance components were estimated by REML for each trait considering year and sites as fixed effects. Broad-sense heritability ($H^2$) was estimated in phenotype data as $H^2 = \frac{\sigma_g^2}{\sigma_g^2 + \sigma_e^2}$, where $\sigma_g^2$ is the genotypic variance and $\sigma_e^2$ is the variance of residuals. Narrow-sense heritability ($h^2$) of each trait was estimated using **G** as the additive genetic variance divided by the total phenotypic variance. Variance components were estimated in R using the package lme4 [46]. The traits were analysed statistically for the estimation of genetic and phenotypic correlation. Bivariate analysis was carried out in ASReml. The fixed effects for bivariate analysis were same as considered in the univariate analysis.

## 2.7. Evaluation of Genomic Prediction

Prediction accuracy was assessed using a five-fold cross-validation (CV) approach where accessions were randomly assigned to folds. Four subsets were combined and formed the training set for estimating the genetic effects. The remaining subset formed the prediction/validation set in which predictions derived from the training set were tested. CV was repeated 10 times to obtain accurate estimates of the average prediction correlation and its standard deviation. Prediction accuracy was measured as the Pearson's correlation between GEBVs and phenotypes, corrected for fixed effects.

Three CV schemes for a single trait and two schemes for multi-trait models were implemented. CV0 was basic cross-validation using complete field trials, where the objective was to predict accessions which have not been phenotyped but genotyped. In CV1, leaving one site out, the goal was to predict rust resistance, where the individuals have not been observed and phenotyped in certain sites. In CV2, accessions were not phenotyped in all years and the main interest was to predict rust resistance for future years in breeding program (leaving one year out).

CV3 and CV4 were multi-trait approaches in which the validation set was either field or greenhouse records. CV3 was set up to predict accessions from field/greenhouse when all phenotypes of the validation set from were excluded from the test population. The aim of CV4 was to assess prediction ability by adding early greenhouse phenotypic data to predict field performance, and thus greenhouse records from the test population were included in the training. This tests a situation common in breeding the program early stages of breeding where lines only have enough seed available for greenhouse assays and this information can be included to predict their future performance in the field. To investigate the efficiency of different prediction models in predicting APR and seedling responses, five models (GBLUP, Bayes A, B, C and LASSO, and RKHS) were included in the CV0 scheme [47]. CV1-CV4 schemes were analysed using GBLUP model in ASreml. Bayesian models and RKHS were fit BGLR-R package [48] with a total of 50,000 cycles of iterations, discarding the first 10,000 samples as burn-in.

## 3. Results

### 3.1. Population Structure

Principle component analysis (PCA), performed using 16,483 SNPs was used to investigate genetic relatedness among the 391 tetraploid wheat accessions. While no clear genetic grouping was detected among accessions, there was evidence of strong admixture as revealed by principal component analysis with two first components jointly explaining 53.2% of the genetic variation (Figure 2). Accessions from Africa and Middle East were loosely clustered by the first PC, while accessions from the Europe showed no clear pattern of clustering.

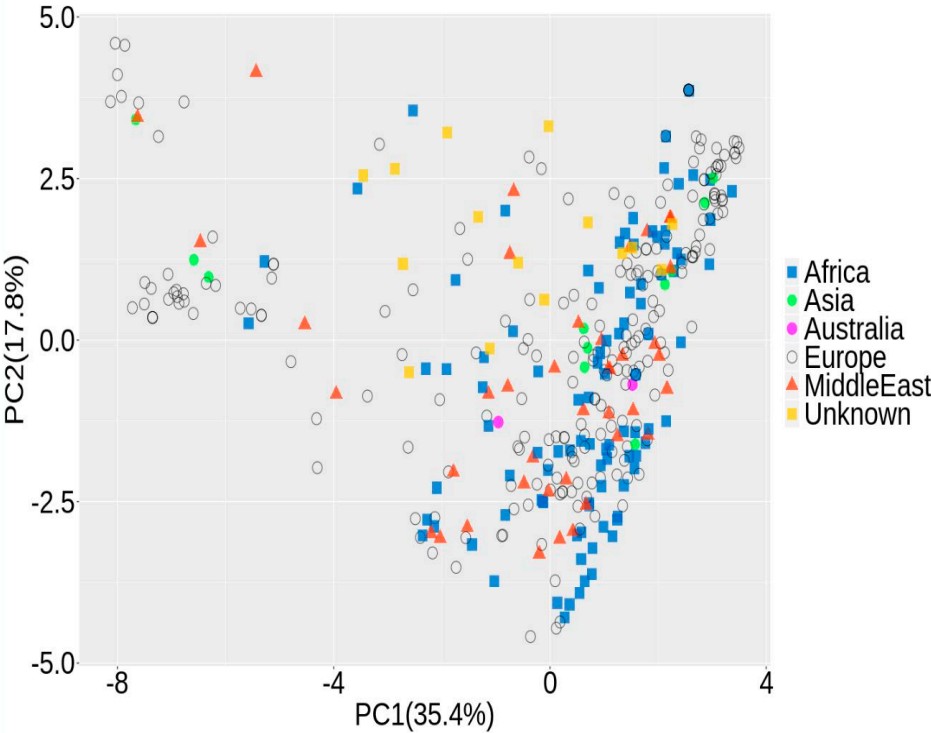

**Figure 2.** Scatterplot of accession distribution based on the first two principal components (PC1, PC2), labelled by origin.

When clustered by species, durum wheat was separated on the first principle component from the other two species, while dicoccon and turgidum were successfully separated on the second principal component (Figure 3).

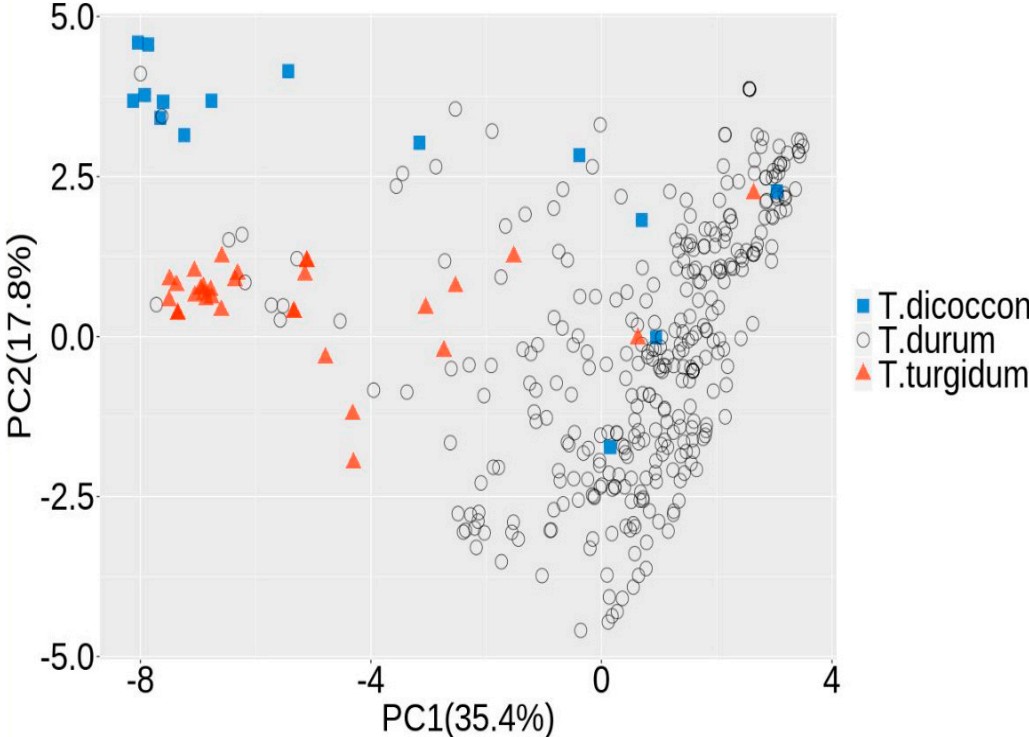

**Figure 3.** Scatterplot of accession distribution based on the first two principal components (PC1, PC2), labelled by species.

Figure 4 illustrates the heatmap of the genomic (**G**) matrix ordered by origin of accessions. Visual assessment revealed small numbers of moderately related accessions in each country, but within-country relatedness did not show a remarkable difference over across-country relatedness. Accessions from Europe comprise small subgroups of related population for Greece and Russia, although the remaining accessions showed no clear relationships (Figure 4).

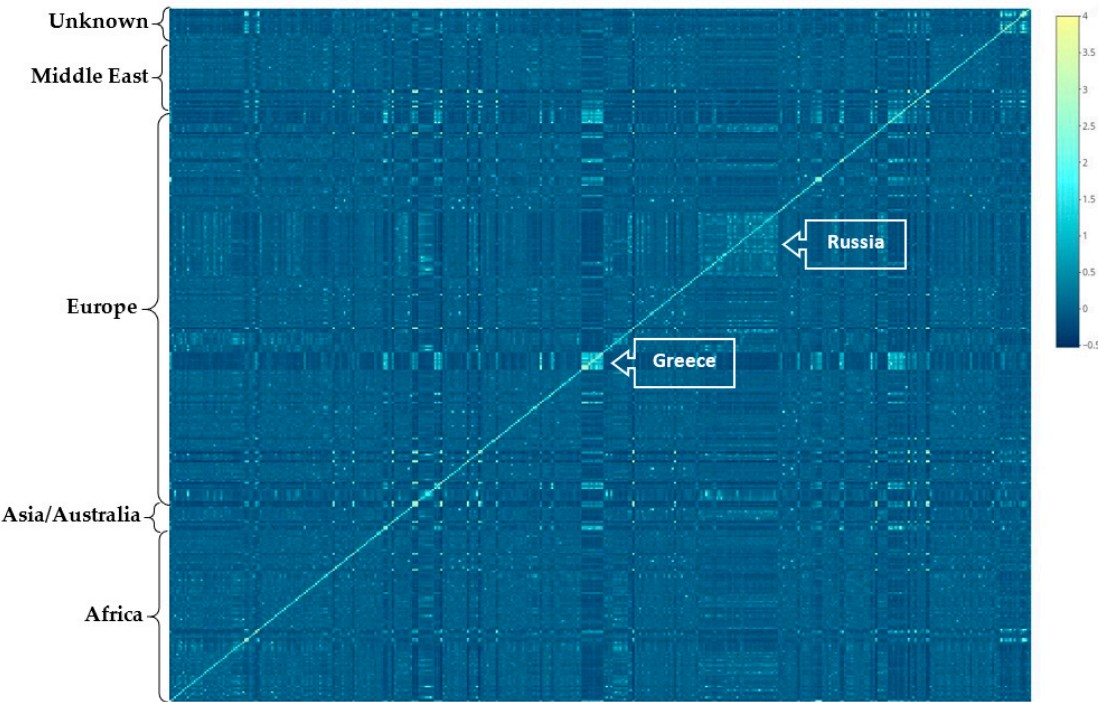

**Figure 4.** Heat map of the genomic relationship matrix (**G**) for tetraploid wheat accessions sorted by country of origin.

## 3.2. Phenotypic Variability and Heritability

Field evaluations revealed quantitative distributions of rust responses among accessions to each of the three rust diseases across both sites and years. Mean severity scores for leaf rust and stripe rust were lower, ranging from 2.5 to 3.5 for leaf rust and 2 to 4 for stripe rut across years and sites, indicating resistance responses of accessions. Stem rust showed minor changes across years and sites with an average severity score of 5 (Figure 5a,b). Heritability of the field traits was calculated at 0.56, 0.42 and 0.56 for leaf rust, stem rust and stripe rust with an additive heritability at 0.33, 0.30 and 0.46, respectively.

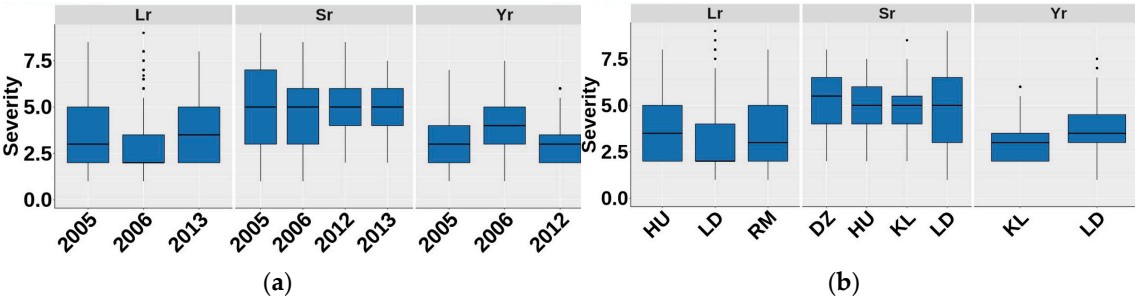

**Figure 5.** Disease severity scores across: (**a**) years for data collected from sites in that year, and (**b**) sites for data collected across years in that site, where 0 is resistant and 9 is susceptible. Sites are HorseUnit (HU), Lansdowne (LD), Richmond (RM), Debre Zeit (DZ), and Karalee (KL).

Seedling stage heritability was estimated at 0.43, 0.56 and 0.39 for leaf rust, stem rust and stripe rust, respectively (Table 2).

**Table 2.** Genetic and additive heritability, prediction accuracy (±) standard errors for adult plant (Field) and seedling resistance (Greenhouse) using different prediction models for rust traits.

| | Field | | | Greenhouse | | |
|---|---|---|---|---|---|---|
| | **Lr** | **Sr** | **Yr** | **Lr** | **Sr** | **Yr** |
| Broad sense Heritability | 0.56 | 0.42 | 0.56 | 0.43 | 0.56 | 0.39 |
| Narrow sense Heritability | 0.33 | 0. 30 | 0.46 | 0.30 | 0.42 | 0.26 |
| GBLUP | 0.70 ± 0.06 | 0.49 ± 0.04 | 0.35 ± 0.03 | 0.53 ± 0.08 | 0.44 ± 0.16 | 0.51 ± 0.09 |
| BayesA | 0.71 ± 0.05 | 0.50 ± 0.08 | 0.35 ± 0.09 | 0.52 ± 0.1 | 0.45 ± 0.15 | 0.52 ± 0.08 |
| BayesB | 0.71 ± 0.07 | 0.50 ± 0.08 | 0.37 ± 0.09 | 0.52 ± 0.08 | 0.43 ± 0.19 | 0.53 ± 0.07 |
| BayesC | 0.71 ± 0.07 | 0.50 ± 0.08 | 0.36 ± 0.08 | 0.51 ± 0.09 | 0.45 ± 0.17 | 0.51 ± 0.07 |
| BayesLASSO | 0.71 ± 0.05 | 0.50 ± 0.08 | 0.36 ± 0.08 | 0.51 ± 0.07 | 0.46 ± 0.14 | 0.51 ± 0.09 |
| RKHS | 0.71 ± 0.06 | 0.50 ± 0.03 | 0.38 ± 0.05 | - | - | - |

*3.3. Phenotypic and Genetic Correlations*

Phenotypic correlations between different environments (sites and years) showed a positive relationship between environments for all traits (Table 3). The associations between stripe rust were significant for trials in Lansdowne and Karalee. Stem rust trials in Australia showed moderate to high phenotypic correlations, but no significant relationship was found between these trials and the experiment in EIAR, Debre Zeit, Ethiopia, presumably due to the presence of different races in each country. Leaf rust also showed a similar trend with moderate to high phenotypic correlations for trials.

**Table 3.** Phenotypic correlations among field trials.

| | | **LD-05** | **LD_06** | | |
|---|---|---|---|---|---|
| **Yr** | **LD_06** | 0.52 * | | | |
| | **KL-12** | 0.47 * | 0.51 * | | |
| | | **LD-05** | **LD-06** | **KL-12** | **DZ-12** |
| | **LD-06** | 0.46 * | | | |
| **Sr** | **KL-12** | 0.43 * | 0.63 * | | |
| | **HU-13** | 0.43 * | 0.62 * | 0.73 * | |
| | **DZ-12** | 0.28 ns | 0.24 ns | 0.22 ns | 0.29 ns |
| | | **LD-05** | **RM-05** | **LD-06** | |
| | **RM-05** | 0.43 * | | | |
| **Lr** | **LD-06** | 0.67 * | 0.50 * | | |
| | **HU-13** | 0.63 * | 0.41 * | 0.72 * | |

ns = not significantly different from zero, * = significant at $p < 0.001$. HorseUnit (HU), Lansdowne (LD), Richmond (RM), Debre Zeit (DZ), and Karalee (KL).

The genetic correlation of APR and the greenhouse data were also analysed. While no clear genetic correlation was revealed between Lr, Sr and Yr APR, there were moderate to high positive associations between APR and ASR of all traits (Table 4).

**Table 4.** Genetic correlations (±) standard errors between field and greenhouse data.

|  | Lr | Sr | Yr | LrGH | SrGH |
|---|---|---|---|---|---|
| **Sr** | 0.20 ± 0.09 |  |  |  |  |
| **Yr** | 0.32 ± 0.10 | −0.07 ± 0.13 |  |  |  |
| **LrGH** [1] | 0.45 ± 0.10 | 0.22 ± 0.13 | 0.24 ± 0.15 |  |  |
| **SrGH** | −0.01 ± 0.13 | 0.65 ± 0.08 | −0.15 ± 0.16 | 0.10 ± 0.13 |  |
| **YrGH** | −0.09 ± 0.11 | −0.25 ± 0.13 | 0.50 ± 0.12 | −0.09 ± 0.14 | −0.10 ± 0.20 |

[1] GH indicates greenhouse data.

### 3.4. Genomic Prediction

Prediction accuracies for single trait analysis of Lr, Sr and Yr APR and seedling resistance (CV0) are shown in Table 2. Bayesian models (A, B, C, and LASSO), GBLUP, and RKHS were used to predict selection accuracies. Predictions ranged from 0.35 (field stripe rust) to 0.71 (field leaf rust) in different models. All methods resulted in very similar accuracies with insignificant differences.

### 3.5. Genomic Prediction Across Years and Sites

The potential of predicting responses in each individual site (CV1) or year (CV2) using data available from other sites or years was tested (Table 5). Mean accuracies for traits ranged from 0.56–0.71 for Lr and 0.29–0.65 for Sr. The least diversity for accuracy was observed for Yr (0.51–0.56) across sites and years. Predictions of Sr APR showed more diversity in one-site/year-out scenarios compared to random predictions, while predictions for Lr performed similarly to random predictions, Yr showed higher prediction across sites and years compared to random predictions.

**Table 5.** Prediction accuracy for adult plant resistance in site-out and year-out validation scenarios.

|  |  | Lr | Sr | Yr |
|---|---|---|---|---|
| Site | Debre Zeit | - | 0.62 | - |
|  | Lansdowne | 0.71 | 0.59 | 0.51 |
|  | Karalee | - | 0.29 | 0.51 |
|  | Richmond | 0.56 | - | - |
|  | Horse Unit | 0.66 | 0.65 | - |
| Year | 2005 | 0.63 | 0.47 | 0.52 |
|  | 2006 | 0.69 | 0.57 | 0.56 |
|  | 2012 | - | 0.58 | 0.51 |
|  | 2013 | 0.66 | 0.65 | - |

The heritability for each site, using data from the predicted site, varied from 0.27 (Richmond) to 0.64 in Horse Unit for Lr, 0.25 (Karalee) to 0.7 (Horse Unit) for Sr, and 0.30 to 0.50 in Lansdowne and Karalee, respectively for Yr confirming the impact of genotype × environment (G × E). Variable heritability estimates reflect the observed variability in accuracy for genomic prediction of the traits. Diverse heritability across sites for Sr is a representative of variable accuracies for this trait.

As expected, regions with similar environments, including climate or races, had lower genotype × environment (G × E), and higher accuracy of models in predicting GEBV across environments. Reduction in accuracy paralleled the decrease in similarities between the years or sites for which models were estimated and validated.

### 3.6. Bivariate Genomic Prediction

Following observing the positive genetic correlations between APR and seedling ASR resistance, bivariate genomic prediction was performed using field data to estimate seedling resistance responses

and vice-versa (ASR, APR). Results showed that accuracies of prediction for both stem rust and leaf rust were higher when field data were used as training compared to greenhouse data.

Prediction accuracy for field responses improved considerably when greenhouse phenotypes of validation set were included in the reference population (CV4) compared to CV3 scenario where ASR phenotypes of individuals in the validation set were removed completely. The accuracy of predicting APR including ASR phenotypes of validation set in the reference population was improved from 0.56 to 0.61 for Lr, 0.38 to 0.48 for Sr and from 0.23 to 0.39 in Yr (Figure 6). This accuracy reflected the genetic correlations between seedling and adult plant responses.

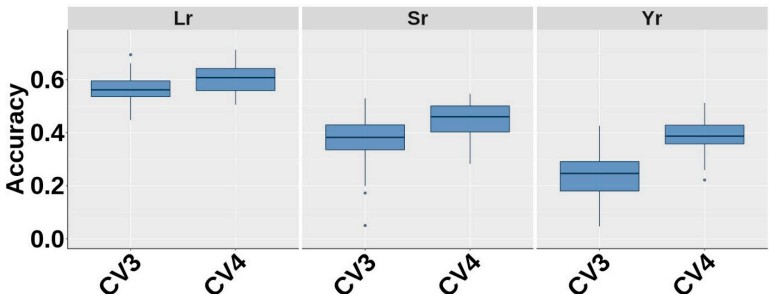

**Figure 6.** Prediction accuracies for bivariate analysis of adult plant resistance (APR), where APR was the target trait and all stage resistance (ASR), was the second trait. CV3: Phenotypes of accessions in validation set were removed entirely from reference population; CV4: respective phenotypes for validation accessions for the second trait were included in the reference population.

## 4. Discussion

Historically, breeders have relied on phenotypic evaluation/selection for disease resistance largely confirmed by ASR genes. In the recent decades, marker assisted selection has been implemented in several breeding programs facilitating the ease of selection for specific genes where diagnostic markers have been developed. GS extends marker-assisted selection to a genome-wide scale; it helps breeders to make more accurate and informed breeding decisions for quantitative traits by estimating the whole spectrum of small to large gene effects, thus taking full advantage of the revolution that molecular markers have brought to crop improvement. Several studies have shown the efficacy of GS in plant breeding for disease resistance [15–24,48–51]. Our study demonstrates the potential for GS in breeding rust resistant tetraploid wheat cultivars and revealed prediction accuracies comparable to other studies that have evaluated GS for rust resistance in wheat [18,20,21]. While genetic relatedness between individuals in the validation and test sets has been shown to be an important factor for achieving good prediction accuracy [52], we did not observe any clear relationships between the accessions used in our study when grouped based on geographical origin. Nevertheless, enough genetic similarities existed between the globally diverse tetraploid accessions used in our study to enable genomic prediction across the population, demonstrating that GS can also be applied to diverse populations.

Due to the existence of major R genes in our population, we examined the effect of different GS models on the prediction accuracy for severity response to each rust disease. These prediction models differed in their assumptions for the underlying genetic architecture and the advantage of using one method over another is still a matter of debate. Despite the existence of major ASR genes in our dataset, we did not observe major differences between the five GS methods tested. Comparable prediction accuracies regardless of the models used have been reported for several wheat traits [53–58]. Our results also showed that the RKHS model performed similar to GBLUP, which is similar in other studies [21], although several studies have reported that non-parametric models perform better than parametric ones [45,56,59]. Although the effect of trait genetic architecture on the performance of different GS methods has been reported in previous studies [19,52,60,61], it has also being shown that when a trait is controlled by more than 20 quantitative trait loci (QTL), the advantages of heterogeneous shrinkage over homogeneous shrinkage can disappear [62].

### 4.1. Predicting Across Years and Sites

We tested cross-validation schemes to study G × E interaction and the potential of predicting genotype response in similar environments (CV1) or for upcoming years (CV2) in breeding programs. Variation between sites particularly for stem rust with more diverse and environments can reflect changes in environmental conditions, climate and/or pathogenic variation, resulting in G × E interactions. This environmental effect was reflected in the low heritability estimate of the trait in some sites. Overall higher prediction accuracies for phenotypes adjusted for fixed effects despite variability across sites differences confirmed the higher G × E interaction. Less diversity in races and environmental differences across sites and years were observed for stripe rust, resulting in more consistent accuracies considering less diversity in locations and races that were evaluated. It has been shown that prediction accuracy varies significantly depending on the trait genetic architecture and rust pathogen isolate composition [22]. Sukumaran et al. [63] compared various models of G × E interactions in yield component of durum wheat predictions and showed that genomic selection models incorporating G × E interaction show great promise for forward prediction and application in durum wheat breeding to increase genetic gains. Based on our results, it can be concluded that similar race composition across years or sites resulted in higher accuracies and therefore lower G × E interaction [11]. The prediction accuracies reported in this study for predicting across site and years are high enough to discard the lines in a real breeding program [63].

### 4.2. Bivariate Prediction

Rust diseases are among major factors affecting yield losses in wheat producing areas worldwide and therefore are among the major targets in breeding programs. These traits are hard to evaluate especially due to the diversity of rust pathogens. The national approach to control rust diseases in Australia does not exist in many nations and the International Maize and Wheat Improvement Centre (CIMMYT), Mexico focused largely on combining multiple race non-specific APR genes that can enhance durability of wheat varieties. The approach of adding greenhouse data to predict APR responses is useful if the aim is to select for rust resistance traits at a very early in the breeding cycle, which cannot be done as not enough seed is available for field screening. To do this, seedlings with the highest genomic breeding values can be selected. This approach provides a cost-effective way to derive genomic prediction for selecting candidates with reliable accuracy for selection. Jia and Jannink [64] found that low-heritability traits can use information from correlated high-heritability traits and consequently achieve higher prediction accuracy in genomic prediction. Other studies showed the potential of improving the genomic prediction of complex traits by incorporating the information from multiple traits collected throughout breeding programs [65–67], which could assist in speeding up breeding cycles.

It is worthwhile noting that plant breeders often want to reduce the undesirable genetic correlations between traits [68], which is necessary for multivariate genomic selection and that this approach is modelled by directly taking advantage of genetic correlation between the target traits.

The genomic correlation between ASR and APR phenotypes for studied traits was very encouraging and led us to attempt bivariate GS in the germplasm. Our results for applying ASR responses as second trait to predict APR showed moderate to high accuracy for genomic prediction of stem rust, stripe rust and leaf rust. The outcome was positive, especially when all ASR phenotypes of validation set were included in the reference population (CV4). In practice, this means that breeding programs could perform greenhouse testing and genotyping of early stage material to estimate more accurate field APR genomic breeding values. Application of greenhouse assessments can be beneficial in overcoming field evaluation barriers to help breeders select resistant accessions in early generations of the breeding cycle. Further work is needed to understand whether the genetic correlation between field APR and ASR is due to single race-specific gene or a combination of genes. Our results also support the feasibility of applying genomic selection as a cost-effective means to enhance genetic gain in tetraploid wheat and to shorten the generation cycle. An increased prediction ability for the multi-trait models indicates the

potential to attain improved genetic gain in wheat breeding programs through these GS approaches [69]. Hayes et al. [70] proposed removing 50–60% of progeny in early stages of breeding cycle based on multivariate prediction results and then using more accurate prediction to select remaining lines in later breeding program stages, which can be a beneficial strategy for moderate prediction accuracies and applying ASR/APR predictions in breeding programs.

## 5. Conclusions

This study examined methods for genomic prediction of APR and ASR for resistance to rust diseases in this tetraploid wheat panel. Further we integrated ASR and APR phenotypes in bivariate genomic selection basis with the following key findings:

(1) Prediction of across years or site were generally equivalent with random cross-valdiation genomic selection outcomes, although there were effects of environmental and pathotype variation in performance of predictions. This approach allows breeders to breed for new environments and modelling performance of elite lines for the future years.

(2) By taking advantage of genetic correlations between ASR and APR, we showed that bivariate genomic selection improves the prediction accuracy for highly evolving rust pathogen adult plant resistance by using information from seedling stage resistance. It is worth noting that it is necessary to study the relationship between APR and ASR for more reliable decisions in the process of practical plant breeding. Use of correlated traits in bivariate models would be expected to reduce phenotyping costs.

The findings of this study could be potentially applied in plant breeding to achieve early generation selection, to reduce field phenotyping cost.

**Author Contributions:** H.D.D., D.P.d.C., M.J.H., H.B., U.B., S.A., study design and methodology; S.A., data analysis; S.A., D.P.d.C., R.P., M.J.H., J.K., S.B., H.B., U.B., H.D.D., writing; H.D.D., supervision; H.M., A.T., M.C., J.K., S.B., H.B., U.B., phenotyping; M.H. genotyping. All authors have read and agreed to the published version of the manuscript.

**Funding:** This research was funded by Agriculture Victoria, GRDC Australia and Australia Awards program.

**Conflicts of Interest:** The authors declare no conflict of interest.

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
