# Peer review of "Genomic Prediction of Rust Resistance in Tetraploid Wheat under Field and Controlled Environment Conditions"

_agronomy, doi:10.3390/agronomy10111843_

Round 1

Reviewer 1 Report

This is a well-written manuscript on the genomic prediction of rust resistance in tetraploid wheat.

The abstract should contain a summary of the results and should exclude more introduction. The abstract must provide enough information that will allow a reader to understand the key findings of the research.

2.2 Seedling and Field Evaluations- Better to make it concise and write exactly what is done.

Author Response

This is a well-written manuscript on the genomic prediction of rust resistance in tetraploid wheat.

Point 1: The abstract should contain a summary of the results and should exclude more introduction. The abstract must provide enough information that will allow a reader to understand the key findings of the research.

Response 1:  we have reduced the introductory sentences and expanded the key findings in the abstract.

Point 2: 2.2 Seedling and Field Evaluations- Better to make it concise and write exactly what is done.

Response 2: We have revised this section of the manuscript to make it more concise.

Reviewer 2 Report

Manuscript is dedicated to evaluate the accuracy of genomic prediction for stem rust, leaf rust and stripe rust resistance in a tetraploid wheat accessions. 391 tetraploid wheat accessions with diverse origins, derived from a collection assembled by AE Watkins were studied.

There are some major drawbacks of the manuscript:

First, geographical grouping is very uncertain and misleading. Table 1 show some historical countries USSR and Abyssinia, together with Ukraine and Ethiopia. Better to use actual countries where accessions were assembled, there is more detailed information in collection description. The same problem is noticeable in Figure 2, when scatterplot based on continents is shown, and its unclear how authors evaluate accessions from former USSR, because it spanned in Europe and Asia. Same is with Figure 4. If authors want to show geographical distribution in continents, such data is misleading. In my opinion, authors must re-evaluate this data, or exclude it from manuscript, because its irrelevant to study or resistance to rust GS, and its not discussed in discussion.

Second, I think discussion must be expanded, and obtained results compared to other publications, especially 4.1 and 4.2.

Manuscript has many minor errors, in example:

Tables 2, 3, 4 lacks statistical description (significance level, information what is ±)

Conclusions are numbered 1 and 3, 2nd conclusion is missing.  

Some references lack volumes, page numbers, DOI or other information (2, 5, 28, 29, 31, 34, 35, 37, 39, 46, 57, 58, 64).

In general, manuscript looks unfinished, therefore must be reconsidered after major revision.

Author Response

Point 1: First, geographical grouping is very uncertain and misleading. Table 1 show some historical countries USSR and Abyssinia, together with Ukraine and Ethiopia. Better to use actual countries where accessions were assembled, there is more detailed information in collection description. The same problem is noticeable in Figure 2, when scatterplot based on continents is shown, and its unclear how authors evaluate accessions from former USSR, because it spanned in Europe and Asia. Same is with Figure 4. If authors want to show geographical distribution in continents, such data is misleading. In my opinion, authors must re-evaluate this data, or exclude it from manuscript, because its irrelevant to study or resistance to rust GS, and its not discussed in discussion.

Response 1: We agree that the annotation of Former Soviet Union into Europe and Asia makes more sense.  We now present the relevant figures in this manner.  These figures are just to present the diversity of the germplasm used and we think they provide important context for the study.

Point 2: Second, I think discussion must be expanded, and obtained results compared to other publications, especially 4.1 and 4.2.

Response 2: We have expanded these discussion sections to improve comparisons to other studies.

Manuscript has many minor errors, in example:

Point 3 : Tables 2, 3, 4 lacks statistical description (significance level, information what is ±)

Response 3 : Apologies for these oversights. We have now made it clear what these symbols refer to.

Point 4: Conclusions are numbered 1 and 3, 2nd conclusion is missing.

Response 4 There are only two conclusions.  Now correctly numbered.

Point 5: Some references lack volumes, page numbers, DOI or other information (2, 5, 28, 29, 31, 34, 35, 37, 39, 46, 57, 58, 64).

In general, manuscript looks unfinished, therefore must be reconsidered after major revision.

Response 5: we have updated the references.  Some references (5,28) are books and ISBN were added.

Round 2

Reviewer 2 Report

Dear Authors,

Thank you for correction of manuscript according to my comments, and I wish you good luck in your future research.